# Spatial Patterns in Actual Evapotranspiration Climatologies for Europe

Simon Stisen [1,*], Mohsen Soltani [1,2], Gorka Mendiguren [1,3], Henrik Langkilde [4], Monica Garcia [5] and Julian Koch [1]

1   Department of Hydrology, Geological Survey of Denmark and Greenland, 1350 Copenhagen, Denmark;
    Mohsen.Soltani@Wetsus.nl (M.S.); ggo@natmus.dk (G.M.); juko@geus.dk (J.K.)
2   Natural Water Production Theme, European Centre of Excellence for Sustainable Water Technology (Wetsus),
    8911 Leeuwarden, The Netherlands
3   Department of Conservation and Natural Sciences, National Museum of Denmark, 2800 Lyngby, Denmark
4   Department of Geosciences and Natural Resource Management, University of Copenhagen,
    1350 Copenhagen, Denmark; henrik5540@hotmail.com
5   Department of Environmental Engineering, Technical University of Denmark, 2800 Lyngby, Denmark;
    mgarc@env.dtu.dk
*   Correspondence: sst@geus.dk

**Abstract:** Spatial patterns in long-term average evapotranspiration (ET) represent a unique source of information for evaluating the spatial pattern performance of distributed hydrological models on a river basin to continental scale. This kind of model evaluation is getting increased attention, acknowledging the shortcomings of traditional aggregated or timeseries-based evaluations. A variety of satellite remote sensing (RS)-based ET estimates exist, covering a range of methods and resolutions. There is, therefore, a need to evaluate these estimates, not only in terms of temporal performance and similarity, but also in terms of long-term spatial patterns. The current study evaluates four RS-ET estimates at moderate resolution with respect to spatial patterns in comparison to two alternative continental-scale gridded ET estimates (water-balance ET and Budyko). To increase comparability, an empirical correction factor between clear sky and all-weather ET, based on eddy covariance data, is derived, which could be suitable for simple corrections of clear sky estimates. Three RS-ET estimates (MODIS16, TSEB and PT-JPL) and the Budyko method generally display similar spatial patterns both across the European domain (mean SPAEF = 0.41, range 0.25–0.61) and within river basins (mean SPAEF range 0.19–0.38), although the pattern similarity within river basins varies significantly across basins. In contrast, the WB-ET and PML_V2 produced very different spatial patterns. The similarity between different methods ranging over different combinations of water, energy, vegetation and land surface temperature constraints suggests that robust spatial patterns of ET can be achieved by combining several methods.

**Keywords:** evapotranspiration; spatial patterns; remote sensing; Budyko; hydrological modeling



## 1. Introduction

Actual evapotranspiration (ET) is a key hydrological flux that together with precipitation determines the upper limit of the water resources available to sustain human needs and freshwater ecosystems. Despite intense research and methodological development, ET continues to be one of the most difficult hydrological states and fluxes to measure, especially on large spatial scales. State-of-the-art methods such as eddy-covariance (EC) flux measurements are subject to systematic energy balance closure problems [1,2] and represent spatial scales in the order of 100–10,000 $m^2$ over homogeneous sites. At the same time, ET can vary significantly across short distances, depending on water and energy availability, which, again, depends on soil, vegetation, elevation, incident angle, micro-climate, etc. Essentially, ET remains impossible to measure accurately at high spatial resolutions

on large spatial scales, which has turned the focus of such estimations towards satellite remote sensing [3]. Although sensors onboard satellites cannot measure ET directly, satellite data offer the opportunity to measure a range of relevant variables, such as land surface temperature, albedo and vegetation cover. Based on these measurements, several methods, models and algorithms have been developed that estimate ET at the spatial coverage and spatial and temporal resolution of the native satellite data.

The main advantage of satellite-based estimates is the ability to produce high-resolution estimates with large coverage based on a uniform information content, potentially producing spatially unbiased and consistent mapping of ET patterns. Although satellite data can ideally provide estimates with a repeat cycle corresponding to the satellite revisit time, such estimates are rarely available due to cloud cover, as the relevant variables are obtained from optical and thermal satellite sensors. However, many hydrological applications do not necessarily require the high temporal resolution of ET, but will benefit from the unique spatial pattern information obtainable from satellite estimates. One such application, which is of particular interest for the current study, is the calibration and validation of spatial ET patterns simulated by distributed hydrological models. For the evaluation of simulated spatial patterns, consistent long-term average spatial patterns of ET at a moderate spatial resolution are highly valuable [4].

Numerous reviews have described satellite-based methods for ET estimation and classified them according to their basic assumptions [5]. A thorough review and classification will not be attempted here; however, to facilitate the subsequent comparisons, a distinction will be made between methods that include surface temperature (thermal wavelenghts) data and methods that are mainly driven by information on vegetation dynamics (red and near-infrared wavelengths). Most moderate-resolution (approx. 1 km) applications of local to regional coverage rely on thermal data [6–12], whereas most continental to global-scale satellite-based ET products are primarily vegetation driven [13–16]. Perhaps one noteworthy exemption from this generalization is the ALEXI model applied from regional to continental scales and driven primarily by the thermal signal [17], although operating on a 5 km scale. The current study will be based on computationally demanding thermal-based ET methods on a continental scale and compare them to vegetation-driven estimates available from global remote-sensing-based ET products.

The main motivation behind this study is to analyze the consistency in spatial patterns of ET from different remote-sensing-based estimates and evaluate their robustness and potential applicability for evaluation of spatial ET patterns, e.g., simulated by distributed hydrological models on the river basin to continental scale. Process-based distributed hydrological models have traditionally been calibrated and validated against river discharge with limited constraint on other hydrological variables. In particular, there has been a lack of attention on the simulated spatial patterns of relevant states and fluxes. Several recent studies have addressed this limitation by including remote-sensing-based spatial patterns in the hydrological model evaluation [18–21]. Due to the increased focus on spatial pattern observations for hydrological model evaluation, there is a need to analyze the consistency and reliability of spatial patterns derived from remote-sensing-based estimates. It has been a common assumption that even though the satellite-based estimates might be uncertain at the point scale, when evaluated against EC measurements or lysimeters, they contain unique and valuable information on spatial patterns [19,22].

The current study aims to compare and validate a set of spatially consistent 1 km resolution ET climatologies over Central Europe using both thermal- and vegetation-driven methods based on MODIS data. The goal of this study is to evaluate the spatial pattern consistency of long-term average ET among the remote-sensing-based estimates at a moderate resolution (1 km) and compare them to alternative ET estimates derived from Budyko [23,24] and water-balance (WB-ET) [25] approaches at a resolution of 25 km. Spatial pattern consistency is investigated within river basins and across Europe in order to evaluate their suitability for informing distributed hydrological models. Previous continental scale benchmarks have focused on North America [26,27] and Africa [28] or

only included vegetation-based RS estimates at a coarser resolution for Europe [29]. A central distinction between the remote-sensing-based methods and the Budyko and WB-ET methods are the way they are constrained either by available energy only (RS methods) or by available water (Budyko and WB-ET). This distinction is fundamental since it determines the upper limit of ET and can result in differences in magnitude of ET. However, our analysis will mainly focus on the spatial pattern similarities on various scales and less on the absolute ET values. This focus originates from recent studies on remote sensing applications for distributed hydrological model evaluation suggested to specifically exploit the pattern information in remote sensing estimates through bias-insensitive performance metrics [4,20,30].

## 2. Data and Methodology

With the eventual goal of evaluating the suitability of RS-based ET estimates for the spatial pattern evaluation of basin-scale hydrological models, the data and methods have been selected accordingly. We focus on long-term moderate-resolution spatial patterns, not on biases or temporal dynamics. The temporal dynamics of RS-ET can also hold valuable information for constraining hydrological models [31], but initially we seek to investigate the consistency of their long-term spatial patterns. We deliberately avoid comparing the RS-ET estimates to point measurements from eddy covariance (EC) towers, since these are very rarely compatible to the spatial scale of the RS estimate and are mostly suitable for assessing the temporal dynamics. However, we do employ EC data from 30 sites to derive an all-weather correction factor for RS estimates that are limited to clear-sky conditions. Consequently, our benchmarks are alternative methods to estimate large-scale ET patterns, namely, the WB-ET and Budyko approaches. These methods are mostly suited to estimate long-term averages and are, therefore, highly relevant to our analysis. In addition, the comparison to benchmark estimates targets large European river basins and their internal spatial patterns of ET. The selected river basins are the 23 largest river basins across Europe with drainage areas above 20,000 km$^2$ and located below 60°N (Figure 1). The intercomparison of remote-sensing-based ET estimates at a 1 km resolution is conducted on the entire land phase of the domain in Figure 1. Based on the data availability, the aggregation period for the study area is 2002–2014.

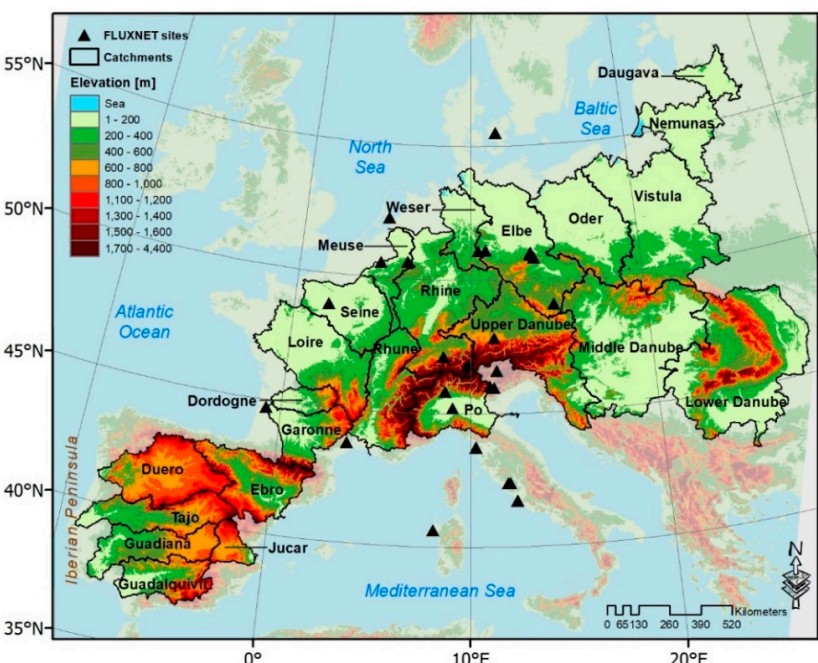

**Figure 1.** The study area showing the distributions of 23 selected catchments and 30 FLUXNET sites with the underlying topography across Europe.

### 2.1. Benchmark Actual Evapotranspiration

For comparison, two benchmark ET estimates, based on other methods and data than the RS ET, are used. These benchmarks are not validation data per say, since they are themselves uncertain and limited. However, they are added as common references to compare and discuss the spatial patterns in the RS ET estimates.

### 2.1.1. Water-Balance ET Approach

The water-balance ET method is a simple calculation of the residual between long-term precipitation P and runoff Q. It assumes that with sufficient temporal aggregation, other terms can be assumed to be insignificant and the water balance reduces to: ET = P−Q. The approach is commonly used for evaluating large-scale ET patterns [25,32] and for evaluating RS-ET products [22,33]. At higher spatial resolutions, groundwater flow patterns can become significant and might require modifications to the approach [34,35]. The WB-ET method is typically based on gridded precipitation data and observed river runoff at the basin outlet [28]. The dependency on observed river runoff poses a major challenge for consistent comparisons across continental scales, due to the lack of consistent records of river runoff for hundreds of basins within a specific time window. To circumvent this challenge, the current study draws on an observation-based European-scale gridded runoff dataset, E-Run [36]. E-Run is established by combining a machine learning approach (Random Forest) with quality-controlled records from 2771 river runoff stations across Europe covering the period 1950–2015. The E-Run pan-European gridded runoff estimate is available monthly at 0.5° resolution. The use of a gridded and temporally complete dataset of runoff is very attractive for the continental-scale WB-ET application in this study since it circumvents the obstacles of complete observational runoff records and coverage of all river basins. The pan-European E-Obs [37] precipitation dataset is also available with a spatial coverage identical to E-Run, but at 0.25° resolution and daily time step. The E-Obs precipitation dataset is the result of a European initiative to collect precipitation records from national meteorological agencies and harmonize them in a pan-European open access dataset.

For the current study, both the E-Run and E-Obs data are aggregated to long-term annual averages for the period 2002–2014 and subset to the coverage in Figure 2 at a 25 km resolution (E-Run was resampled using bilinear interpolation). The resulting long-term precipitation and runoff maps are illustrated in Figure 2.

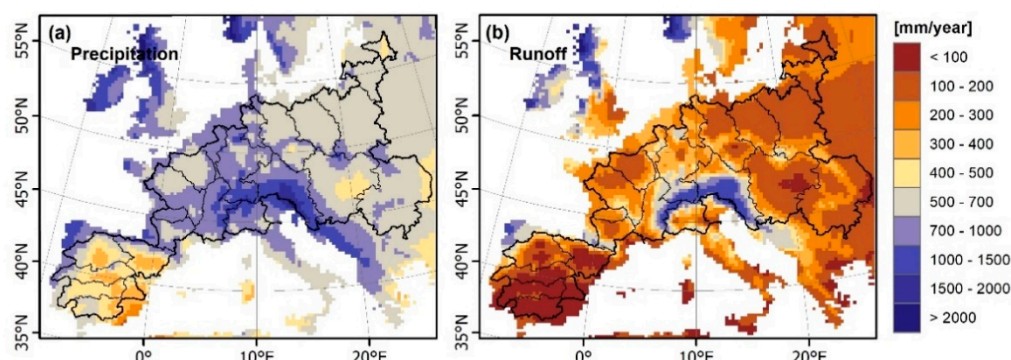

**Figure 2.** The water-balance ET approach components at a 25 km spatial resolution: (**a**) E-Obs precipitation and (**b**) E-Run surface runoff, both for the period of 2002–2014.

### 2.1.2. Budyko ET Approach

The Budyko approach [23] describes how the ratio of atmospheric water supply (i.e., precipitation P) and climatic water demand (i.e., potential evaporation PET) drives the partitioning of P into evapotranspiration ET and streamflow Q on catchment scales [38], as shown in Figure 3. In this approach, which postulates a phase of transformation of green water into water vapor, ET reflects not only the partitioning of water but also the

partitioning of radiant energy [39]. Thus, ET is limited by water in dry conditions and by energy in wet conditions [40]. Therefore, the Budyko curve can be used to evaluate the performance of different ET products, as it provides a reference condition for the water balance. Using the Budyko method, previous studies have demonstrated that there is a strong relationship between the climate aridity, i.e., PET/P and the evaporative index, i.e., ET/P (and runoff ratio) in many catchments globally [41–45]. Further, some studies have also attempted to describe a high capability of the aridity index in controlling the partitioning of precipitation into Q and ET [46–48].

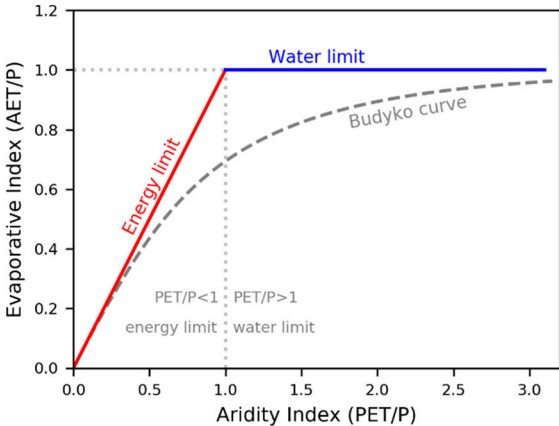

**Figure 3.** The Budyko curve representing the energy-limit (ET = PET) and the water-limit (ET = P) lines.

In this study, we calculated ET/P and PET/P for each of the catchments and plot the results against the Budyko curve using the Equation (1) (for the period 2002–2012), as follows [23]:

$$\left[ \frac{PET}{P} \tanh\left( \frac{1}{\frac{PET}{P}} \right) \left( 1 - \exp^{-\frac{PET}{P}} \right) \right]^{0.5} \tag{1}$$

The advantage of using the Budyko function is that meteorological data, P and PET (unlike Q) are more readily available. We also applied an empirical equation in the Budyko framework to calculate the grid-based ET (for the period 2002–2012), as follows [49]:

$$ET[mm/year] = f(P, PET) = \frac{P}{\left( \left( \frac{P}{PET} \right)^n + 1 \right)^{1/n}} \tag{2}$$

where *n* is a dimensionless parameter that modifies the partitioning of P into ET and Q [50]; in other words, it represents the integrated effects of catchment attributes (e.g., vegetation, soil, climate, topography and seasonality) and human activities on P partitioning, with a common value of *n* = 2.6 as a best fit to the original Budyko curve [51–53], and applicable to different catchments. However, in this study, we obtained a fully distributed *n* value derived from NDVI across the study area using empirical equations proposed by [54], accounting for the vegetation component of catchment attributes. For this, the MOD13A2 global 16-day NDVI at a 1 km spatial resolution was used to represent the vegetation characteristics of the land surface over Europe; then, it was resampled to 25 km and the mean annual NDVI climatology (2002–2012) was obtained from the dataset. The vegetation coverage was normalized (NDVInorm) using the minimum and maximum NDVI values of 0.05 and 0.8, respectively, as follows [55]:

$$NDVI_{norm}[-] = \frac{NDVI - NDVI_{min}}{NDVI_{max} - NDVI_{min}} \tag{3}$$

Finally, the catchment-gridded $n$ values were achieved (see Figure 4b) using the following equation, which is a linear regression-based $n$ parameterization between the catchment-gridded $n$ and the long-term normalized annual vegetation coverage [54]:

$$n[-] = a \times \mathrm{NDVI}_{\mathrm{norm}} \times b \tag{4}$$

where $a$ and $b$ are constants with the values of 2.36 and 1.16, respectively. For further detailed review of the Budyko-type equations and applications of the approach on gridded data and comparison to satellite ET, the reader is referred to, e.g., [24,52,54,56–58].

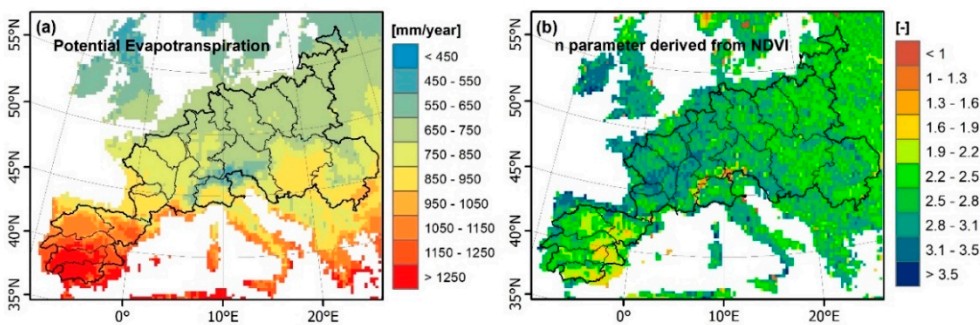

**Figure 4.** The Budyko ET approach components at a 25 km spatial resolution: (**a**) potential evapotranspiration and (**b**) distributed n parameter values derived from NDVI, both for the period of 2002–2012.

We also used gridded average PET estimates (represented in Figure 4a), as input for Equations (1) and (2). These PET products are globally available at daily 0.05° resolution for the period 1979–2012, and freely accessible at https://wci.earth2observe.eu/ (accessed 1 June 2020). The data used here are from the three products using the Penman–Monteith, Priestley–Taylor and Hargreaves approaches [59], from which averages were calculated at both a 25 km grid resolution (used in Equation (2)) and for each basin (used in Equation (1)). The precipitation data (see Figure 2a) are the same as those used in the WB-ET approach (see Section 2.1.1).

## 2.2. Remote-Sensing ET Datasets

Satellite remote-sensing approaches are suitable to provide spatially distributed ET information. However, ET cannot be measured directly from space, but several algorithms are available to estimate ET by means of remotely sensed techniques [22]. Four remote-sensing ET estimates are considered in this study all based on data from the Moderate-Resolution Imaging Spectroradiometer (MODIS). With a spatial resolution of approx. 1 km, such ET estimates can provide information on a relevant scale for Europe, which is suitable for the calibration and evaluation of distributed process-based hydrological and land-surface models with a focus on spatial patterns of ET. Therefore, remote-sensing-based ET models such as GLEAM [16] or the Meteosat-based LSA SAF ET [15] were not considered in this study as they do not provide the desired resolution to assess spatial patterns within catchments. In addition, given the focus on observed spatial patterns of ET, the analysis of RS-ET is confined to approaches that are derived directly from satellite observations without use of a water budget model. All RS-ET datasets were resampled to 1 km grid cells, and first aggregated to monthly climatologies and then to annual timescales over the period 2002–2014 (see Figure 1). Missing data in the monthly climatologies, which are exclusively found in the winter months, were gap filled using the minimum value for the given cell before calculating the annual mean. The four remote-sensing ET algorithms used in this study are briefly described below, and further detailed characteristics of the RS-ET products are given in Table 1. Two of the estimates, the vegetation-based MODIS 16 and PML_V2 are readily available globally and easy to access. The two other estimates, PT-JPL$_{\mathrm{thermal}}$ and TSEB, are in addition to vegetation, based on daily land surface temperature data,

which are not readily available for download. PT-JPL$_{thermal}$ and TSEB require extensive preprocessing, especially on the European scale, since daily cloud masking and corrections for acquisition time is necessary. Moreover, the thermal-based ET methods are limited to clear sky conditions and need to be adjusted before comparison to the other ET estimates, which reflect all weather conditions. This will be elaborated in Section 2.2.5.

**Table 1.** Characteristics of remote-sensing ET products. P-M: Penman–Monteith; GPP: Gross Primary Production; P-T: Priestly–Taylor; FPAR: Fraction of Photosynthetically Active Radiation; GMAO: Global Modeling and Assimilation Office; and GLDAS: Global Land Data Assimilation System.

| Product | Spatial Resolution | Temporal Resolution | ET Algorithm | Input Data Sources | References |
|---|---|---|---|---|---|
| MODIS16 | 1 km | 8-day | PM | MODIS (land cover type2, FPAR/LAI, albedo) and flux towers /GMAO (forcing data) | [13] |
| PML_V2 | 500 m | 8-day | PML [1] | MODIS (LAI, albedo, emissivity) and GLDAS (forcing data) | [14] |
| PT-JPL$_{Thermal}$ | 1 km | Daily | PT | MODIS (LST, emissivity, albedo, LAI, FPAR, NDVI), E-OBS (air temperature) ERA-Interim reanalysis (forcing data) | [11] |
| TSEB | 1 km | daily | TSEB (based on PT) | MODIS (LST, albedo, LAI) and ERA-Interim reanalysis (forcing data) | [9] |

[1] ET and GPP coupled through surface conductance in PM.

### 2.2.1. MODIS 16 Evapotranspiration (MOD16)

The global MODIS 16 ET product [13] is originally based on the Penman–Monteith (PM) equation [60]; however, the model has been modified [61] to account for parameters not readily available from space [62]. The algorithm considers both the surface energy partitioning and environmental constraints on ET, and it uses ground-based meteorological observations and remote sensing data from MODIS to estimate evapotranspiration. As a vegetation-based ET model, the MODIS16 ET applies leaf area index (LAI) as a scalar for estimating canopy conductance and the Normalized Difference Vegetation Index (NDVI) for calculation of the vegetation cover fraction [13,61].

### 2.2.2. Penman–Monteith–Leuning Evapotranspiration (PML_V2)

The global PML_V2 ET product [14] is a coupled diagnostic biophysical model. Based on the Penman–Monteith–Leuning equation (PML), it utilizes various MODIS-derived variables, such as LAI, albedo and emissivity, together with GLDAS meteorological forcing data, as model inputs. Due to the water and carbon cycle coupling, the algorithm considers the carbon constraint on water flux estimates at a high spatial resolution (500 m) globally; therefore, PML_V2 is suitable for assessing the influence of carbon-induced impacts on the terrestrial ET [14].

### 2.2.3. Priestly–Taylor Jet Propulsion Lab Thermal Evapotranspiration (PT-JPL$_{Thermal}$)

In the PT-JPL family of models, actual evapotranspiration is estimated on a daily time scale relying on the Priestley–Taylor (PT) evapotranspiration equation. ET is partitioned into soil evaporation and canopy transpiration for each pixel assuming a layered canopy approach. The first step is to estimate surface net radiation. The shortwave net radiation is estimated based on the albedo and incoming shortwave radiation. The longwave net radiation is calculated by applying Stefan–Boltzman's law for the land and the atmosphere, requiring land and air temperatures and land and air emissivities, respectively. Afterwards, the soil net radiation flux can be estimated using a Beer-Lambert law where the radiation flux reaching the soil depends on the canopy LAI and a *fixed* extinction coefficient.

To move from potential to actual values, empirical multipliers between 0 and 1 reflecting stomatal regulation and/or surface conductance are estimated, similarly to Jarvis type stomatal conductance. These multipliers or biophysical constraints describe responses in conductance to variations in soil moisture, temperature, canopy greenness, and canopy moisture levels. The green canopy fraction ($f_g$) relies on the ratio between absorbed and

intercepted PAR while the plant moisture constraint ($f_M$) estimates the departure of the $f_{APAR}$ at a given day from the maximum $f_{APAR}$ within the whole time series.

The PT-JPL$_{Thermal}$ method implemented here is based on the model developments described in [11,63], especially adjusted for arid regions. Compared to the original PT-JPL version [64], the PT-JPL$_{Thermal}$ version implemented here estimates the soil moisture constraint ($f_{sm}$) based on the concept of apparent thermal inertia, where the diurnal oscillation of land surface temperature (LST) should scale with the level of surface dryness as well as the surface albedo instead of the atmospheric dryness. This approach has been proven to work better than the original one in water limited and Mediterranean systems [63]. A thermal inertia approach is also now implemented as part of the ECOSTRESS PT-JPL ET product [65].

The European-scale PT-JPL$_{Thermal}$ model was setup using mainly MODIS-derived input data, such as daytime and nighttime LST, emissivity, NDVI, LAI, albedo and FPAR. Meteorological data was derived from the ECMWF ERA Interim dataset and average daily air temperature was obtained from the E-OBS dataset at 0.25 deg resolution resampled to the MODIS resolution of 1 km. In the result section we will refer to PT-JPL$_{Thermal}$ simply as PT-JPL in description, figures and tables.

### 2.2.4. Two-Source Energy Balance Evapotranspiration (TSEB)

The Two-Source-Energy-Balance (TSEB) model used in the current study is based on an implementation of the method proposed by [9]. The model belongs to the two-layer type of models, that treat soil and canopy separately and estimates instantaneous fluxes of sensible (H) and latent heat (LE) on the basis of remotely sensed LST, albedo and vegetation parameters in combination with climate variables of radiation, wind speed and air temperature. The TSEB model by [9], includes an iterative step that adjusts the Priestly Taylor constant $\alpha_{PT}$ until the total energy balance is satisfied, meaning that Rn = LE + H + G, while LE$_{soil}$ and LE$_{canopy}$ > 0. Our implementation follows the description in [66] and is based on the code provided by the pyTSEB package. The model is run for all days in the period 2002–2014 for all clear-sky pixels across the study domain (Figure 1). The required input data are similar to the PT-JPL$_{Thermal}$ model described above and include radiative forcings, LST, albedo, LAI, and fraction of green vegetation. The fraction of green vegetation $F_g$ is based on a method proposed in [66], which discriminates between a greening and a senescence phase of the annual vegetation development, where the $F_g$ is higher relative to the NDVI in the greening phase compared to the senescence phase.

### 2.2.5. Adjustments to Improve Comparability between RS-ET Estimates

To increase comparability between the four RS-ET estimates, a set of processing steps have been conducted. The datasets are all subset and spatially aggregated to the same domain and grid size (1 km), while a common mask at a 1 km resolution is applied, representing the urban and snow-covered areas, according to the MODIS landcover classification.

The thermal-based estimates from PT-JPL and TSEB initially represent instantaneous estimates at the time of satellite overpass. Therefore, only overpass times corresponding to the window 12–5-14 h local solar time have been utilized. These are assumed to represent the peak in land surface temperature at around 13 h. Correspondingly, the radiative forcing data from ECMWF required to run the PT-JPL and TSEB models have been generated by a gridbased weighting scheme between the 9, 12 and 15 h UTC ECMWF data. This will remove forcing biases across the large domain originating from differences in local solar noon (UTC-time). Estimates of instantaneous latent heat (LE) in W/m$^2$ at 13 h are converted to daily LE (MJ/m$^2$/day) and subsequently to ET (mm/day). The conversion is based on dependencies on latitude, day of year and time of day according to the formulations by [67].

Finally, the PTJPL and TSEB estimates represent clear sky estimates in contrast to MODIS16 and PML_V2, which, like the Budyko and WB-ET estimates, represent all-

weather conditions. Although the current study remains focused on the spatial pattern comparison of the different ET estimates, an effort has been put on adjusting the clear sky estimates to all-day estimates. This is done by examining a large dataset of observed ET timeseries from the FLUXNET database across Europe. In total, 30 EC stations with consistent daily LE data records for the period 2002–2014 were selected. For these data, two long-term average monthly LE averages were calculated. One using all days available and another using only the days where PT-JPL and TSEB estimates were available (clear sky conditions) for the given pixel of each FLUXNET station. This analysis, illustrated in Figure 5, revealed that clear sky days consistently overestimate LE compared to all-day estimates across all months and by around 13% on average (ratio of 0.88). This analysis is based solely on the observed LE data, using satellite data only to distinguish clear sky days (days where a MODIS LST value is available). The factor of 0.88 is subsequently used to scale the PT-JPL and TSEB clear sky estimates to all-day estimates more comparable to the other ET estimates.

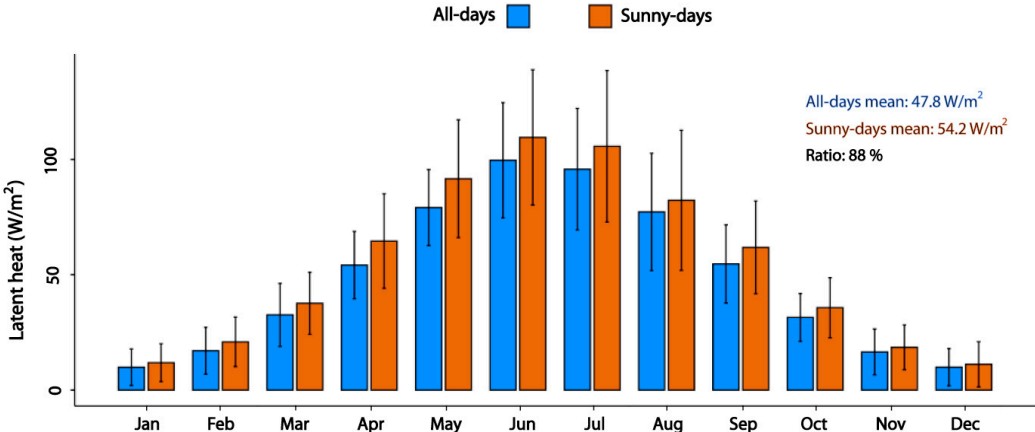

**Figure 5.** The long-term monthly mean (2002–2014) latent heat flux (LE) during all days and sunny days for 30 FLUXNET stations across Europe. See Figure 1 for the spatial distribution of the EC sites.

### *2.3. ET Spatial Pattern Evaluations*

#### 2.3.1. Spatial Efficiency Metric SPAEF

The spatial efficiency metric (SPAEF) is a spatial pattern similarity evaluation metric initially developed to compare the spatial pattern performance of distributed hydrological models, e.g., in comparison to satellite-derived spatial patterns of key hydrological states and fluxes [4,30]. The SPAEF metric builds upon the idea and structure of the Kling-Gupta Efficiency (KGE) [68], with two modifications. The SPAEF aims to be bias insensitive and focus specifically on the pattern. This is achieved by reformulating the standard deviation (σ) ratio term of KGE to a ratio (β) of coefficients of variation and by substitution of the bias term of KGE with a histogram matching term (γ). The SPAEF, with an optimal value of 1, will be used throughout the comparisons of ET patterns, both across Europe and within river basins.

$$\text{SPAEF} = 1 - \sqrt{(\alpha - 1)^2 + (\beta - 1)^2 + (\gamma - 1)^2} \tag{5}$$

$$\alpha = \rho(A, B) \text{ and,} \tag{6}$$

$$\beta = \left( \frac{\sigma_A}{\mu_A} \right) \Big/ \left( \frac{\sigma_B}{\mu_B} \right) \text{ and,} \tag{7}$$

$$\gamma = \frac{\sum_{j=1}^{n} \min(K_j, L_j)}{\sum_{j=1}^{n} K_j} \tag{8}$$

2.3.2. Estimation of Correlation Structures Using the Copula Approach

The empirical Copula function [69] was employed to estimate the underlying dependence structures amongst the ET products applied in this study. The Copula approach states that any multivariate distribution function can be decomposed into the marginal distributions and a Copula, and the dependence structure (i.e., true relationships) between variables is robustly estimated [70]. Using the asymmetrical-based Copula distributions, it becomes feasible to describe which parts of the datasets are highly or weakly correlated, i.e., at lower-tails, upper-tails, or the middle range of a joint distributions. Here, to estimate the Copula density $C$, first $\hat{F}_x$, $\hat{F}_y$ of the marginal distributions, was obtained using the empirical distribution function as an estimator, then the pseudo-observations $(\hat{u}, \hat{v}) = (\hat{F}_x(x), \hat{F}_y(y))$ were defined [71]. The Copula density, which is purely based on the real data [72], was finally estimated as the joint density of $(\hat{u}, \hat{v})$. If $\{r_1(1), \ldots, r_1(n)\}$ and $\{r_2(1), \ldots, r_2(n)\}$ denote marginal distribution-based rank space values, the empirical Copula was then defined as:

$$C_n(u, v) = 1/n \sum_{t=1}^{n} \mathbf{1}\left( \frac{r_1(t)}{n} \leq u, \ \frac{r_2(t)}{n} \leq v \right) \tag{9}$$

where $u = F_x(x)$, $v = F_y(y)$ and $\mathbf{1}(\ldots)$ is the indicator function that takes a value of 1 if the argument $(\ldots)$ is true [73], and $n$ is the sample size. Further information about Copulas can be found in, e.g., [70,74].

2.3.3. Hierarchical Cluster Analysis

To assess the overall similarity between the ET products and rank their spatial pattern variability across the individual catchments considered in this study, a hierarchical cluster analysis was performed using the built-in *R heatmaply* function [75]. To enable comparability of the ET products from different approaches/algorithms, the data were normalized before application of the cluster analysis. Here, the catchment annual mean ET values for all products were used.

**3. Results**

*3.1. Spatial Patterns of Benchmark ET*

The spatial distributions of long-term annual mean evapotranspiration estimated from the water-balance approach (WB-ET) and the empirically based Budyko approach (Budyko ET) are illustrated in Figure 6a,b, respectively. The two estimates of ET vary considerably, both regarding magnitude and spatial pattern, despite utilizing the same precipitation data. The Budyko ET (Figure 6b) displays a spatial pattern with higher ET values in Central Europe, lower values in Eastern Europe and sharp gradients on the Iberian Peninsula. The WB-ET (Figure 6a) displays generally lower ET in Central Europe, very low ET in the Alps, Scotland and South of Norway and less gradients in the Iberian Peninsula. The Budyko approach represents a smoother and more seamless ET spatial pattern (Figure 6b) compared to WB-ET (Figure 6a), and seems to follow more closely the expected climatological patterns across Europe.



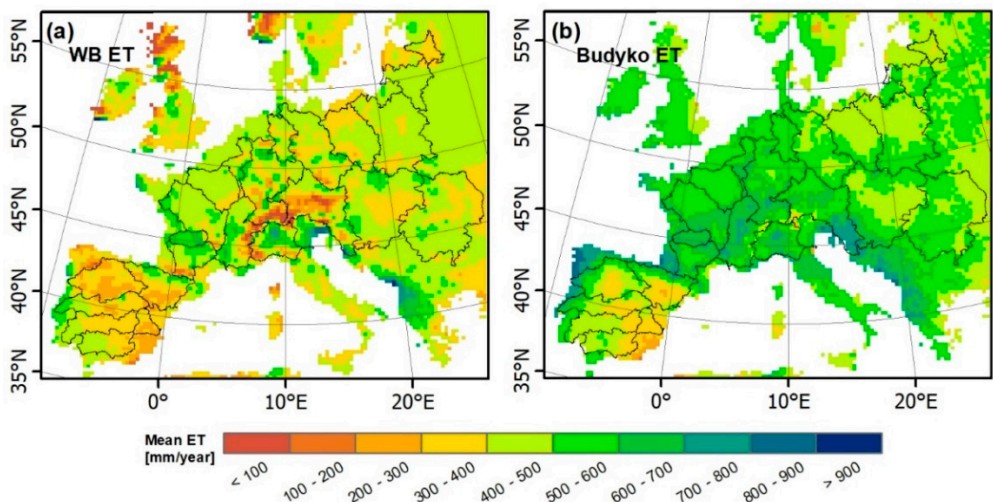

**Figure 6.** The grid-based (25 km) annual mean benchmark ET spatial patterns over the period of 2002–2014: (**a**) water-balance WB-ET and (**b**) Budyko ET. The catchments boundaries (N: 23) are overlaid on the ET maps. See Figure 1 for detailed descriptions.

### 3.2. Spatial Patterns of Remote-Sensing ET

Figure 7 presents the high-resolution (1 km) long-term average actual evapotranspiration products derived from the remote-sensing models of MODIS16 (Figure 7a), PML_V2 (Figure 7b), PT-JPL (Figure 7c) and TSEB (Figure 7d). Initially, it can be observed that there are some general resemblances between the spatial patterns of three of the products, MODIS16, PT-JPL and TSEB, on a continental scale, whereas PML_V2 stands out with a markedly different pattern. The PML_V2 exhibits a dominant North-South pattern, with the lowest values in the North, supposedly due to a strong control by radiation forcings on the ET pattern. When quantifying the spatial pattern similarity pairwise, through the SPAEF metric (Table 2), any pairing with PML_V2 ranks lowest with values between 0.11 and 0.20. Although the MODIS16, PT-JPL and TSEB patterns show a general agreement across Europe, there are differences, most notably, MODIS16 has relatively lower ET on the Iberian Peninsula, TSEB has lower values in Eastern Europe and PT-JPL has higher values in Central Europe. Amongst the RS-ET maps, the spatial patterns of TSEB vs. PT-JPL and MODIS16 vs. PT-JPL show the highest spatial pattern similarity (SPAEF: 0.67 and 0.5, respectively), while MODIS16 vs. TSEB has SPAEF: 0.28 (Table 2). Besides the spatial pattern comparisons, absolute values of each ET map can be compared; here, PML_V2 ET (Figure 7b) and PT-JPL ET (Figure 7c) show the lowest (~200–700 mm/year) and the highest (~400–900 mm/year) ET rates, respectively.

**Table 2.** The SPAEF calculation between the RS-ET products with a 1 km spatial resolution at full domain.

| ET Products | SPAEF | ET Products | SPAEF |
|---|---|---|---|
| [PML_V2, MODIS16] | 0.12 | [TSEB, MODIS16] | 0.28 |
| [PTJPL, PML_V2] | 0.20 | [TSEB, PML_V2] | 0.11 |
| [PTJPL, MODIS16] | 0.50 | [TSEB, PTJPL] | 0.67 |

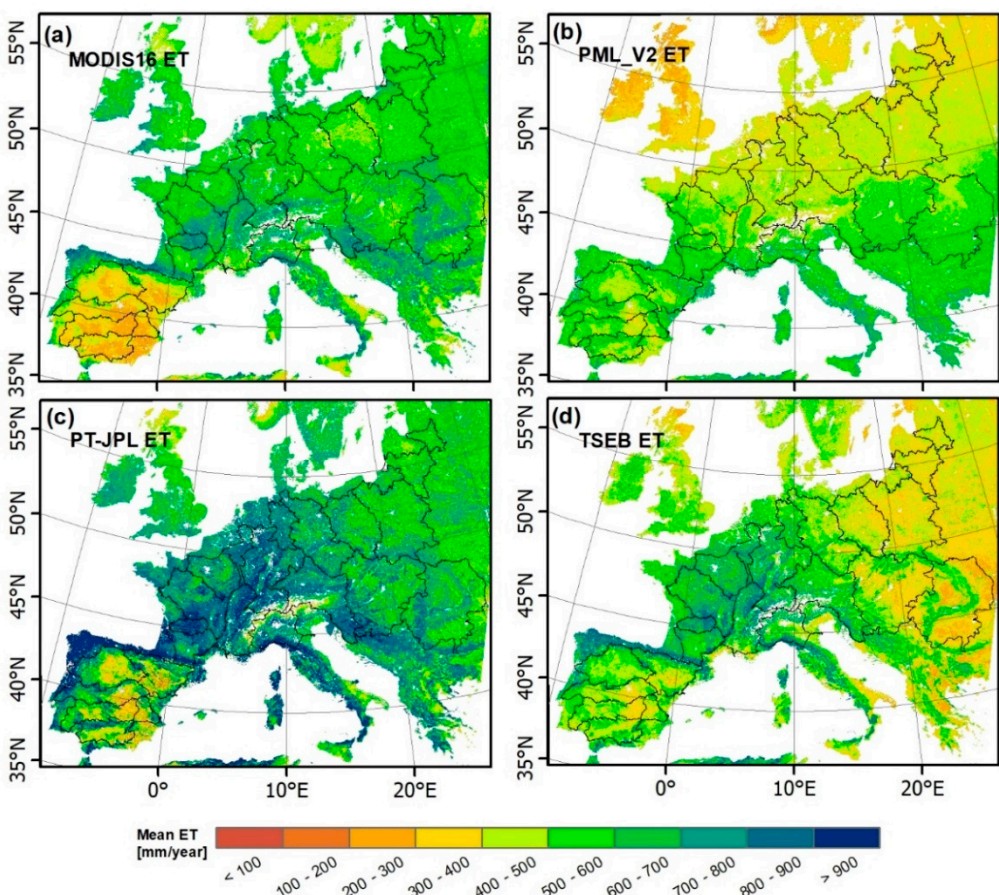

**Figure 7.** The grid-based (1 km) annual mean RS-ET spatial patterns over the period of 2002–2014: (**a**) MOD16, (**b**) PML_V2, (**c**) PT-JPL, and (**d**) TSEB. The catchments boundaries (N: 23) are overlaid on the RS-ET maps. See Figure 1 for detailed descriptions.

Subsequently, the RS-ET spatial patterns are compared on a river basin scale, by SPAEF calculated for the internal spatial patterns within the 23 selected basins, Figure 8. As for the continental scale comparison, PML_V2 displays the lowest similarity to the other RS-ET estimates. In many cases, PML_V2 indicates a strongly negative correlation when compared with other RS-ET maps. For example, a very different spatial pattern is identified between PML_V2 and TSEB (SPAEF: −1.0) in the Lower Danube basin (Figure 8). Among the three other estimates, TSEB vs. PT-JPL has the highest similarity (average SPAEF 0.38) followed by TSEB vs. MODIS16 (average SPAEF 0.34). This indicates that although TSEB and MODIS16 showed some differences in the cross-continental comparison due to systematic regional differences, the within-basin patterns can be quite similar. In contrast, MODIS16 vs. PT-JPL, which had the highest similarity in the cross-continental comparison, has a somewhat lower pattern similarity within each river basin (average SPAEF 0.19). As shown in Figure 8, SPAEF between individual catchments in TSEB ET and PT-JPL ET range from minimum: 0.12 in Lower Danube and maximum: 0.64 in Loire (see Figure 1 for the location of the catchments). Likewise, the spatial efficiency also indicates high correlations for most of the catchments between MODIS16 and TSEB ET and a wider range with a maximum SPAEF value of 0.71 obtained in Ebro catchment and minimum at Weser with −0.10. With regard to MODIS16 and PT-JPL, the numbers of catchments with a high SPAEF are limited (Figure 8).

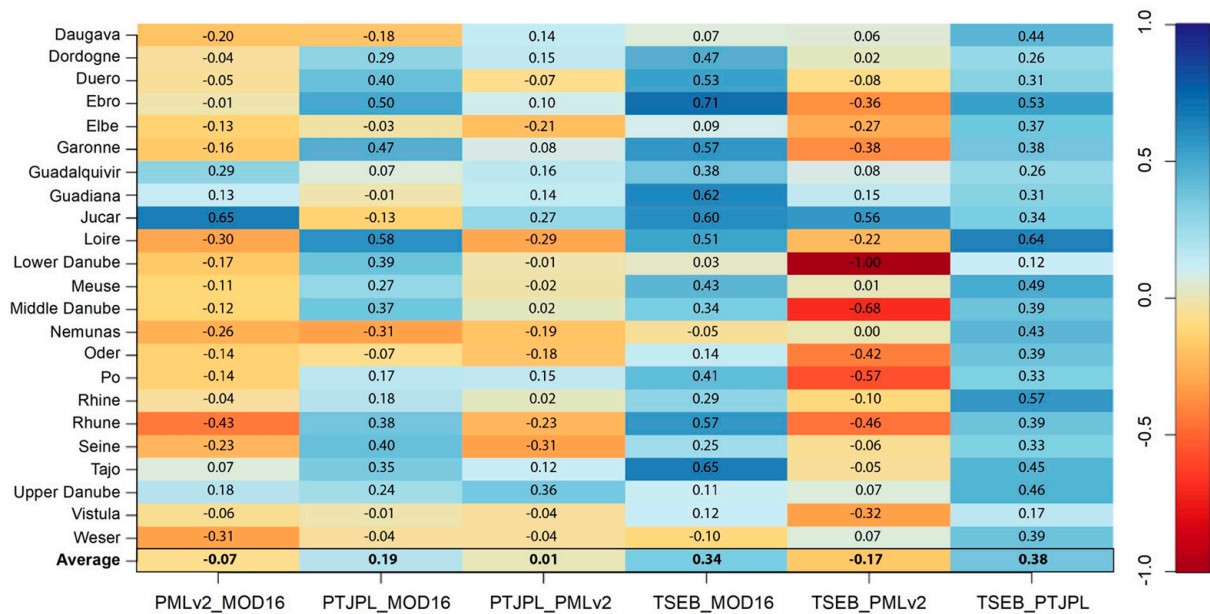

**Figure 8.** The catchment-based SPAEF calculation between the RS-ET products with a 1 km spatial resolution over the period of 2002–2014. The catchments are shown in y-axis (rows) and the RS-ET datasets are displayed in x-axis (columns).

### 3.3. Benchmark ET Compared with Remote-Sensing ET

The annual mean climatologies (25 km) of the two estimated benchmark ETs (WB and Budyko) and the four remote-sensing-derived ETs (MODIS16, PML_V2, PT-JPL and TSEB) are presented in Figure 9. For a better comparison of the spatial patterns and calculation of statistics, the ET maps are normalized (each ET map is divided by its own mean value) and RS-ET estimates are aggregated to 25 km. This analysis focuses on the 23 river basins, because the WB and Budyko approaches are most commonly applied at the river basin level. The spatial distribution of the WB-ET (Figure 9a) does not represent a seamless pattern, but appears patchy, whereas the Budyko ET (Figure 9b) spatial pattern shows a smoother transition from lower ET in the south and eastern parts of the Iberian Peninsula, transitioning to higher values in a quite homogeneous Central Europe, to intermediate values in Eastern Europe. This general trend across Europe also appears in the three RS-ET maps, MODIS16 (Figure 9c), PT-JPL (Figure 9e) and TSEB (Figure 9f), which all exhibit a strong similarity to the Budyko estimate (Figure 9b). Again, the PML_V2 estimate (Figure 9d) shows a very different spatial pattern. Statistically, the highest SPAEF values between benchmark ETs and RS ETs are obtained between Budyko ET and MODIS16 ET (SPAEF: 0.43) and PT-JPL ET (SPAEF: 0.4), whereas the corresponding SPAEF values against the WB-ET are 0.17 and 0.07, respectively (Table 3). Similarly, a notable difference is found between the two benchmark ET estimates when compared to the TSEB data; here, a negative SPAEF (−0.11) is obtained between WB-ET and TSEB ET (Figure 9a,f), while the corresponding SPAEF value is 0.25 for Budyko ET vs. TSEB ET (Figure 9b,f). Amongst the remote-sensing-based ET maps, the spatial patterns of PT-JPL ET vs. TSEB ET and PT-JPL ET vs. MODIS16 ET indicate the highest similarity (SPAEF: 0.61 and 0.53, respectively). On the contrary, the largest spatial discrepancy is observed when PML_V2 ET is compared to other RS-ET maps. These findings are similar to the European-scale comparisons in high resolution (Table 2). Generally, the ET estimates based on the Budyko, MODIS16, TSEB and PT-JPL show the highest similarity, whereas WB-ET and PML_V2 display great dissimilarity.

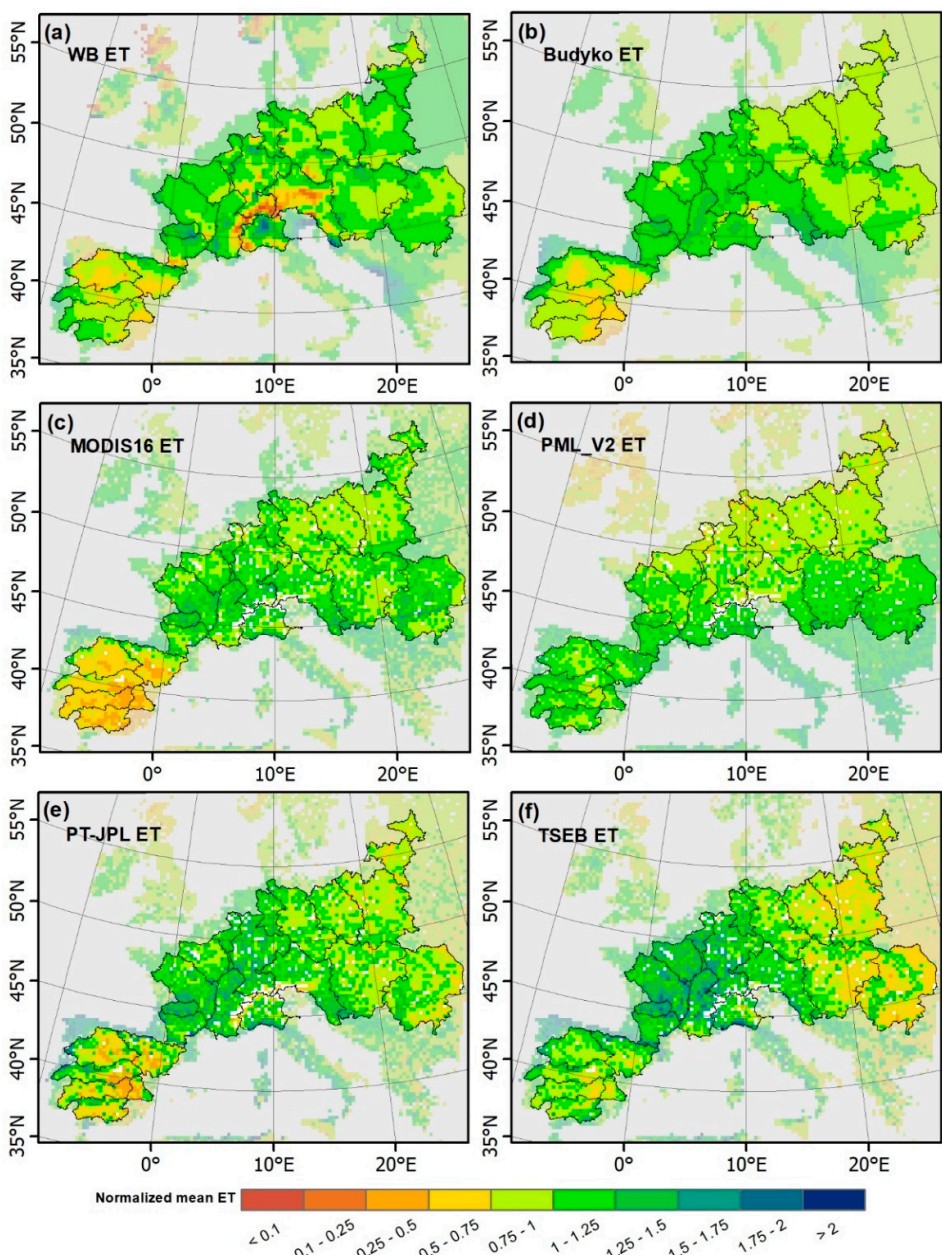

**Figure 9.** The normalized (each ET map was divided by its mean value) catchment-based annual mean ET spatial patterns with a 25 km spatial resolution over the period of 2002–2014: (**a**) WB-ET, (**b**) Budyko ET, (**c**) MODIS16 ET, (**d**) PML_V2 ET, (**e**) PT-JPL ET and (**f**) TSEB ET.

**Table 3.** The SPAEF calculation between the ET products with a 25 km spatial resolution across catchments.

| Data | SPAEF | Data | SPAEF |
|---|---|---|---|
| [WB, Budyko] | 0.26 | | |
| [WB, MODIS16] | 0.17 | [Budyko, MODIS16] | 0.43 |
| [WB, PML_V2] | 0.03 | [Budyko, PML_V2] | 0.15 |
| [WB, PT-JPL] | 0.07 | [Budyko, PT-JPL] | 0.40 |
| [WB, TSEB] | −0.11 | [Budyko, TSEB] | 0.25 |
| [TSEB, MODIS16] | 0.25 | [MODIS16, PT-JPL] | 0.53 |
| [TSEB, PML_V2] | 0.03 | [MODIS16, PML_V2] | 0.05 |
| [TSEB, PT-JPL] | 0.61 | [PT-JPL, PML_V2] | 0.24 |

### 3.4. ET Evaluation Using Budyko Curve Analysis

Figure 10a evaluates the performance of WB/RS-ET products against the Budyko curve, and the corresponding ET products for each of the color-coded catchments are illustrated in Figure 10b. It can be seen that the majority of the catchments in all ET products deviate from the Budyko curve by falling either under or above the curve, showing a tendency to underestimate or overestimate catchment ET, compared to the standard Budyko approach ($n = 2.6$). WB-ET follows the Budyko curve better, compared to the RS-ET products, as most of the catchments fall close to but generally a little under the curve, indicating a systematic difference, as previously observed in the spatial patterns (see Figure 9a,b). In contrast, PT-JPL and TSEB show a clear tendency for higher ET in all catchments (see Figure 9e). It is also noted that the RS-ET products in some cases exceed water-limit and/or energy-limit in calculating ET. This is mainly caused by differences in the available energy used for the different approaches given as PET for the Budyko (PET) and net radiation as calculated with each of the RS-ET models. Besides, in some basins, additional input of water from irrigation (when the water-limit is exceeded) or an additional loss, e.g., through the groundwater system (in case of exceeding the energy-limit), can contribute to these deviations, although these contributions are likely small on the basin scale. In general, the interpretation of Figure 9 should focus on the relative differences between basins along the AET/P axis. Overall, catchments with a higher aridity index (PET/P > 1.5) are mainly located in water-limited environments in Southern Europe. Conversely, catchments with a lower aridity index (PET/P < 1) are mostly located in the energy-limited environments, found in Central Europe (Figure 10b). Overall, our Budyko results (i.e., the aridity index analysis) are consistent with findings of other studies over Europe (e.g., [76]).

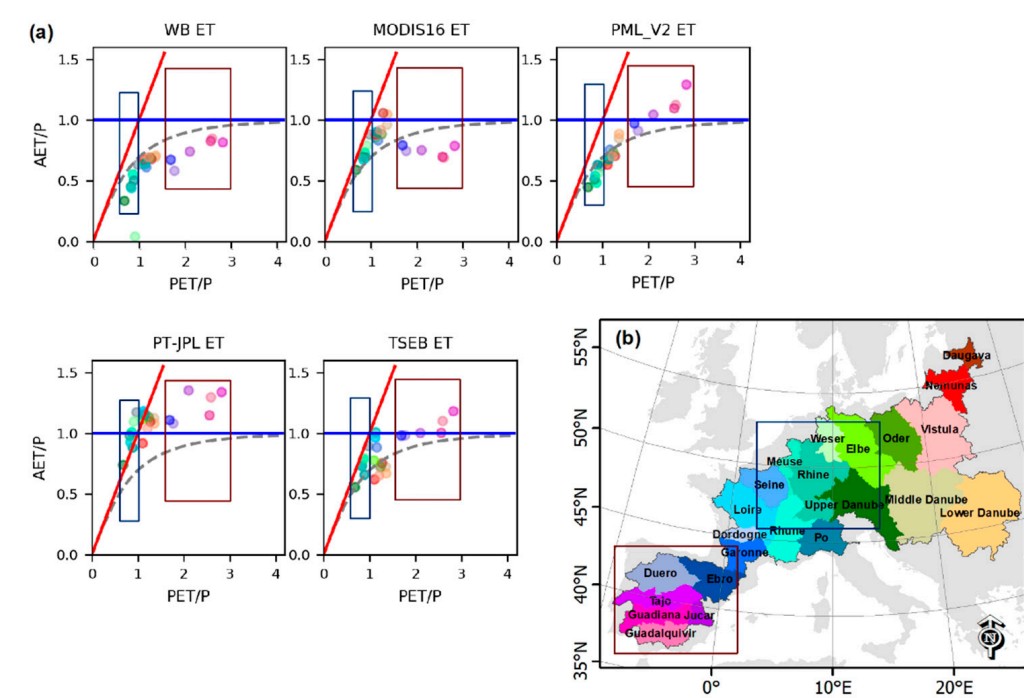

**Figure 10.** (**a**) Evaluation of the long-term (2002–2014) estimated WB-ET and derived RS-ET products using the Budyko curve; (**b**) geographical location of the catchments corresponding to the color of points in panels *a*. The red (blue) box represents catchments with the highest (lowest) aridity index, which fall in the water-limited (energy-limited) environment.

### 3.5. Copula-Based Dependence Structures of ET Products

The empirical Copula densities amongst the four remote-sensing ET products at a 1 km resolution are illustrated in Figure 11, where the underlying dependence structure (correla-

tion) of the joint relationships is observed with strongly varying densities in the different percentiles. The Copula density of MODIS16 ET against other RS-ET datasets indicates that the dependence structure between MODIS16 and PT-JPL is significantly symmetrical with maximum densities in the lower-left and upper-right corners; this means that the highest correlation is found at both low and extreme values (Figure 11b). However, the density between MODIS16 vs. PML_V2 and TSEB is asymmetrical, where the highest dependency is seen in the upper-right corners; this implies that they are strongly concordant in the higher ranks of the distribution only. Moreover, the middle/higher ranges of MODIS16 ET corresponded to minimum values of PML_V2 and TSEB ET products, indicating a bias among them (Figure 11a,c). PML_V2 indicates an insignificant dependency against PT-JPL and TSEB, where the highest density is mainly found in the upper-right corners (high values), and also a negative correlation can be observed for the middle/lower ranges of their distributions (Figure 11d,e). Among the RS-ET products, the most significant symmetric dependence structure with maximum densities in the lower-left and upper-right tails is observed between PT-JPL and TSEB (Figure 11f); they also show a positive correlation for the middle range of values, which was not seen for other RS-ET products applied. The Copula analysis, therefore, implies that the spatial patterns in PT-JPL and TSEB are very similar regardless of their absolute values (see Figure 9e,f), while MODIS16 displays some similarity to PT-JPL in particular. Contrastingly, PML_V2, again, displays a high degree of dissimilarity.

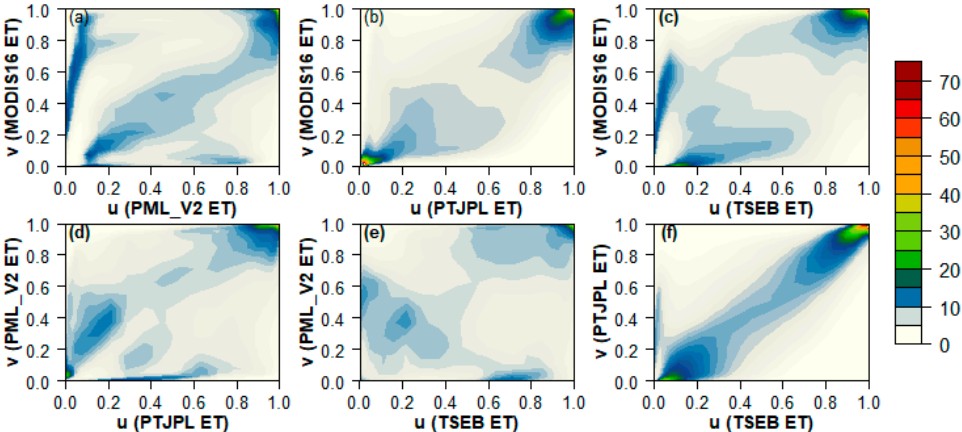

**Figure 11.** The empirical Copula densities (1 km grid at full-domain) amongst the satellite RS-ET products. (**a**–**c**) MODIS16 vs. other RS-ET estimates (**d**,**e**) PML_V2 vs. other RS-ET estimates, and (**f**) PTJPL vs. TSEB. The sample size is 2,247,522 data tuples for the ET datasets.

As shown in Figure 12, in general the empirical Copula density of WB-ET and Budyko ET (as reference ETs) shows a very similar dependence structure against remote-sensing ET products. The Copula density of the reference ETs against MODIS16 and PT-JPL is significantly symmetrical with maximum densities in the lower-left and upper-right corners. This means that the highest correlation is found at the very low and extreme values (Figure 12a,c,e,j); however, the concordance is stronger between Budyko ET and the RS-ET products. In contrast, the density function of the reference ETs against PML_V2 and TSEB (Figure 12b,d,f,h) indicates an insignificant (and partially negative in the lower range of the distributions) asymmetric dependence structure with the maximum density in the upper-right tail only (high values); this is especially evident between Budyko ET and TSEB ET (Figure 12h). Overall, a strong linear relationship can be seen between the reference ETs against MODIS16 and PT-JPL and a weak nonlinear relationship to PML_V2 and TSEB. However, dependence structures are stronger for PT-JPL and TSEB when compared to Budyko than WB-ET. Interestingly, the Copula function captures a significant symmetric dependence structure with maximum densities in the lower-left and upper right corners between WB-ET and Budyko ET approaches (Figure 12i), although there are also relatively

high densities in the lower right corner. This symmetry, presumably caused by the shared precipitation data, is somewhat contrary to the previous comparisons, which revealed quite different spatial patterns. Moreover, a clear systematic bias is observed not only in the lower/higher, but also in the middle range of values between the two reference ETs (unlike the WB/Budyko ETs against the RS-ET products).

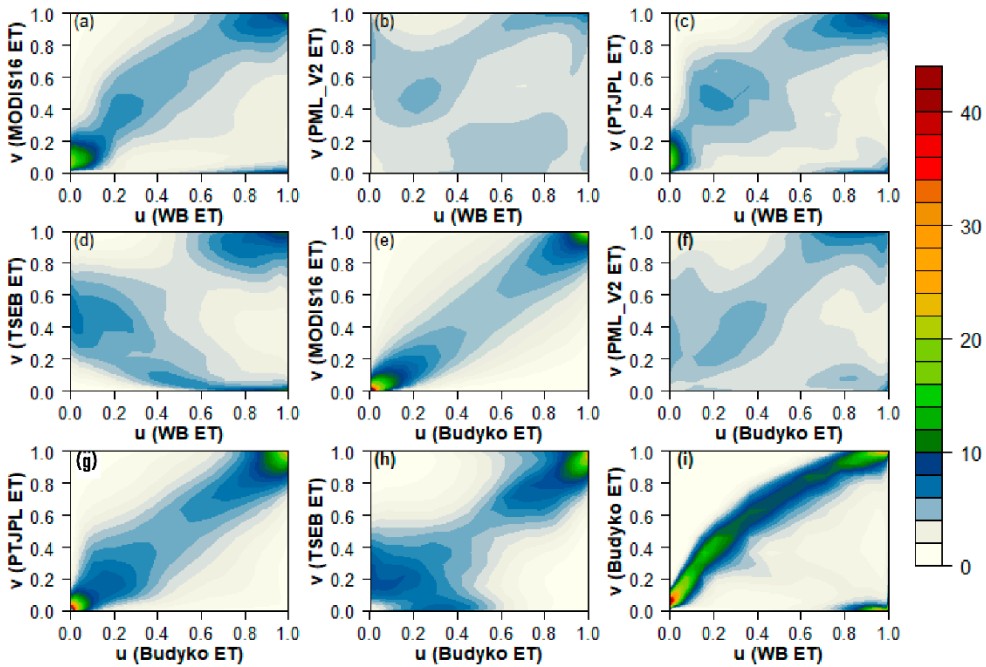

**Figure 12.** The empirical Copula densities (25 km grid at catchment-scale): (**a–d**) WB-ET and (**e–h**) Budyko ET against RS-ET datasets, and (**i**) WB-ET against Budyko ET. The sample size is 3413 data tuples for the ET datasets.

### 3.6. Similarity Assessment Using Cluster Analysis

The six ET products and the selected 23 catchments used herein are classified into three and four main clusters, respectively, based on the ET-cluster (x-axis) and the basin-cluster (y-axis), as shown in Figure 13. The first ET-cluster is formed by the four estimates of PT-JPL and Budyko as well as MODIS16 and WB-ET; the second and the third ET-clusters are represented by TSEB and PML_V2, respectively. Within the first ET-cluster, PT-JPL and Budyko show the highest level of similarity, which is in good agreement with the Copula results (See also Figure 12g) and ET spatial patterns (see Figure 9b,e). TSEB and PML_V2 form individual "clusters", with TSEB being more similar to cluster 1 than PML_V2. In terms of the basin-cluster, catchments located in the south-west central parts of Europe formed the first basin-cluster, where the highest ET values are found (e.g., Dordogne, Garonne and Loire); the second basin-cluster formed with catchments located in the north-central parts, which have the middle ranges of ET values (e.g., Weser, Meuse, and Rhine); the third basin-cluster formed with catchments located in Eastern Europe, where the middle to lower ranges of ET values are found (e.g., Lower Danube, Vistula and Daugava), and finally, the fourth basin-cluster formed with catchments located in the Iberian Peninsula, where the lowest ET values (except for the PML_V2 ET) are observed (e.g., Jucar, Guadalquivir, Tajo). See Figure 1 for the locations of the catchments. It is worth mentioning that these basin-clusters from 1 to 4 (Figure 13) are in very good agreement with the Budyko findings (see Figure 10), in which catchments with the lowest (highest) aridity index fall in the energy-limited (water-limited) environment, respectively.

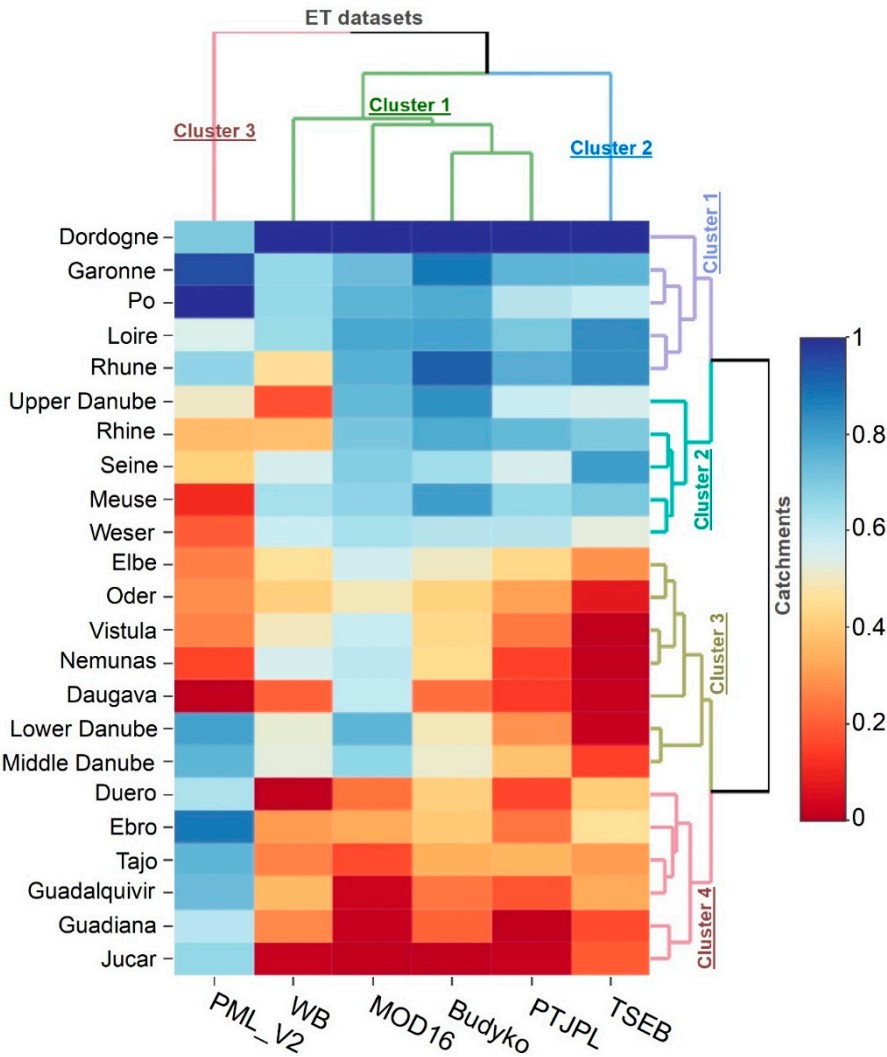

**Figure 13.** The hierarchical cluster analysis for the annual mean (2002–2014) normalized ET datasets in a 25 km grid, representing the overall similarity ranking among the ET products for each of the catchments across the study area. See Figure 1 for the locations of the catchments.

## 4. Discussion

The current study remains focused on the spatial pattern comparison of the long-term average ET estimates. However, it is worth addressing dissimilarity in the general magnitude of ET between some of the ET estimates. The WB-ET seems to underestimate ET compared to the other estimates, while PT-JPL seems to overestimate. The underestimation of WB-ET could originate from the E-Obs precipitation data, but is more likely to be attributed to the overestimation of runoff in the E-Run dataset, which according to [36] has a tendency to overestimate in Central and Northern Europe, where it constitutes a larger fraction of the water balance, and underestimate in Southern Europe, where runoff rates are generally low. The WB-ET approach is obviously very sensitive to the discharge data used, and the approach used here, based on gridded runoff produced by a machine learning approach, seems to be inadequate. Instead, the traditional WB-ET approach based on real observed runoff records within a well-defined catchment boundary would be more appropriate. This is, however, complicated by the absence of continuous and complete runoff records on the European scale. Moreover, the ability to investigate fully distributed, grid-based spatial patterns is hampered using catchment-based runoff. The main controlling factor for the spatial pattern of WB-ET is, however, the precipitation data,

while the vegetation plays only a secondary role through the distributed parametrization of the *n* parameter.

Differences in the general ET levels between the applied methods can also be attributed to differences in available energy input and clear sky vs. all weather estimates. The latter has been handled in the current study by a correction factor derived from ET observations on a point scale and assumed to be applicable for the entire domain. This is a novel approach and although the assumption of general applicability cannot be tested, the approach is able to correct the TSEB estimate to levels that are very comparable to Budyko and MODIS16, whereas the PT-JPL improved by the correction, but still overestimates compared to other datasets. The high ET levels of the PT-JPL estimate could likely originate from differences in the estimation of potential ET or available energy. The empirically derived factor for all-weather conditions (0.88), based on a very large dataset, could be useful as a simple correction method for other studies based on thermal RS-ET estimates. Comparing methods that are constrained by precipitation and energy (Budyko) to methods that are only constrained by energy (RS ET) indicates that patterns and levels are similar, see, e.g., Figure 6b (Budyko) in comparison to Figure 7a (MODIS16) and Figure 77d (TSEB).

It is important to consider two aspects of spatial pattern information for hydrological model evaluation, both the large-scale patterns and the internal catchment patterns. MODIS16, PT-JPL, TSEB and the Budyko show some similarities on the European scale, with the lowest ET values on the Iberian Peninsula, the highest in Central Europe and intermediate in Eastern Europe. WB-ET and PMLv2 do not agree with the general spatial pattern across Europe; although the WB-ET pattern has some similarities, the correlation is hampered by very low ET estimates in high-elevation regions. The controlling factor for the PML_V2 pattern is radiation, with higher values in south and lower values in north, while vegetation and other factors have a limited impact, which is very different from the other estimates.

A major difference between the RS-ET estimates is whether they are driven mainly by vegetation (MODIS16 and PLM-V2) or by a combination of vegetation and land surface temperature (PT-JPL and TSEB). This difference seems to result in a systematic difference in the level of ET in Eastern Europe compared to Central Europe, where the temperature-driven models tend to result in significantly lower ET in Eastern Europe. Similarly, the Budyko approach also resulted in lower ET values in Eastern Europe. These differences are clearest in the normalized maps in Figure 9 and in the catchment clusters (3 and 4) in Figure 13. A recent study by [29] also evaluated a range of RS-ET datasets over Europe and compared them to machine learning approaches. Their study generally showed similar spatial patterns across Europe, especially pronounced similarity to the MODIS16 across Europe, with high ET levels in Eastern Europe. All models in the [29] study belonged to what can be referred to as vegetation-driven models (as MODIS16), without constraints induced by land surface temperature or precipitation.

Regarding the second aspect of spatial pattern evaluation, the within-basin patterns, there are strongest similarities between TSEB and PT-JPL and secondly between TSEB and MODIS16. Even for these RS-ET comparisons, the spatial pattern similarities (Figure 8), do, however, vary greatly, ranging from around 0.0 to 0.6 between basins. This indicates that although all three estimates show similarity and could be used in isolation or in combination to assess the within-basin spatial patterns of hydrological models, there is clear potential for using several RS-ET pattern estimates in an ensemble approach to illuminate uncertainties. Another approach moving forward could be to merge RS-ET and Budyko ET in an ensemble mean or utilize the combined water and energy constrain in Budyko to bias correct RS-ET, or reversely refine the grid scale of the coarser Budyko estimate using moderate-resolution RS-ET estimates. Such merged approaches could prove robust and minimize the uncertainty of selecting a specific RS-ET estimate. Based on the results of this study, it would not be recommended to include either the grid-based WB-ET or the PML_V2 estimates, which differ significantly from the other estimates regarding spatial pattern information.

Our evaluation approach has deliberately avoided comparison to point-scale evapotranspiration measurements. This decision originates from the focus on spatial pattern and the fact that comparison to point-scale measurements is mainly relevant in the evaluation of temporal dynamics. The scale difference and local variability of evapotranspiration does not warrant a spatial pattern evaluation.

## 5. Conclusions

Four ET estimates (MODIS16, TSEB, PT-JPL and Budyko) display a high degree of agreement in spatial patterns across major river basins in continental Europe. Likewise, the within-basin spatial patterns are analyzed across the moderate-resolution RS-ET estimates revealing good agreement, especially between TSEB and PT-JPL and between TSEB and MODIS16, although with a large spread among basins. These four estimates represent both purely energy-constrained (MODIS16, TSEB and PT-JPL) as well as water-and-energy-constrained (Budyko) methods. In addition, the RS-ET methods span algorithms driven mainly by vegetation (MODIS16) and driven by both vegetation and land surface temperature (TSEB and PT-JPL). This indicates that even with fundamental differences in methods, similar spatial patterns in ET are obtained. A way forward could be to merge these four estimates, and other estimates that agree on the general patterns, in a robust ensemble.

The WB-ET estimate based on gridded runoff data based on a machine learning approach seems to be unsuited for the WB-ET approach, since the resulting spatial pattern deviates significantly from the other estimates. The RS-ET estimate PML_V2 has a completely different ET pattern from all other estimates and does not follow the expected variability across Europe. Therefore, PML_V2 seems unsuited as a reference for the long-term spatial pattern of ET for Europe.

MODIS16, PT-JPL, TSEB and the Budyko estimates show common patterns and are all considered valid estimates of spatial patterns in ET on the European scale. Similarly, MODIS16, TSEB and PT-JPL are regarded as suitable datasets for evaluating internal spatial patterns at the river basin level for hydrological model evaluation.

**Author Contributions:** Conceptualization, S.S., M.S. and J.K.; methodology, S.S., M.S., J.K., G.M. and M.G.; software, S.S., M.S., G.M., M.G., H.L. and J.K.; validation, S.S. and M.S. formal analysis, S.S., M.S., J.K., G.M. and H.L.; investigation, S.S., M.S., G.M., M.G., H.L. and J.K.; resources, S.S., M.S., G.M., M.G., H.L. and J.K.; data curation, S.S., M.S., G.M., H.L. and J.K.; writing—original draft preparation, S.S., M.S. and J.K.; writing—review and editing, S.S., M.S., G.M., M.G., H.L. and J.K.; visualization, S.S., M.S. and J.K.; supervision, S.S.; project administration, S.S.; funding acquisition, S.S. All authors have read and agreed to the published version of the manuscript.

**Funding:** This research was funded by the Villum Foundation (http://villumfonden.dk/, accessed on 5 April 2021) through their Young Investigator Programme (grant VKR023443). In addition, funding was obtained through the GEOERA TACTIC project funded from the European Union's Horizon 2020 research and innovation programme under grant agreement No 731166.

**Informed Consent Statement:** Informed consent was obtained from all subjects involved in the study.

**Data Availability Statement:** The gridded long-term average datasets on actual evapotranspiration derived in the study are available (upon publication) at the "Gridded European Evapotranspiration Climatologies", https://doi.org/10.22008/FK2/T6NBHH, accessed on 5 April 2021, GEUS Dataverse repository.

**Acknowledgments:** We acknowledge the E-OBS dataset from the EU-FP6 project UERRA (http://www.uerra.eu; https://www.ecad.eu/download/ensembles/download.php) (accessed 1 June 2020) and the Copernicus Climate Change Service, and the data providers in the ECA&D project (https://www.ecad.eu) (accessed 1 June 2020). PET data are accessed through https://wci.earth2observe.eu/. in 1 June 2020. Likewise, we acknowledge the E-Run dataset provided through the PANGAEA repository, Gudmundsson, Lukas; Seneviratne, Sonia I (2016): E-RUN version 1.1: Observational gridded runoff estimates for Europe, link to data in NetCDF format (69 MB). PANGAEA, https://doi.org/10.1594/PANGAEA.861371 (accessed 1 June 2020). We acknowledge

ECMWF for providing access to the ERA-Interim data and NASA for access to MODIS products. Both MODIS16 and PML_V2 data are accessed through Google Earth Engine. This work benefited from eddy covariance data acquired and shared by the FLUXNET community. We would like to thank Hector Nieto for sharing the pyTSEB package through https://github.com/hectornieto/pyTSEB (accessed 1 January 2018).

**Conflicts of Interest:** The authors declare no conflict of interest.

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
