# Peer review of "Spatial Patterns in Actual Evapotranspiration Climatologies for Europe"

_remotesensing, doi:10.3390/rs13122410_

Round 1
Reviewer 1 Report
A massive work has been done here around AET which is probably the most difficult meteorological variable to measure. I do not have any comment since I found the manuscript almost perfect. I'm pretty sure that will have significant future merit in AET assessment. Well done to the authors!
Author Response
Reviewer 1
A massive work has been done here around AET which is probably the most difficult meteorological variable to measure. I do not have any comment since I found the manuscript almost perfect. I'm pretty sure that will have significant future merit in AET assessment. Well done to the authors!
Reply: we really appreciate the positive review and hope that the manuscript will have an impact.
Reviewer 2 Report
ET is important for Earth's water and energy balance, despite the difficulties in accurate estimating ET. There are many new ET estimates with the emergence of long-term and high-resolution satellite data and auxiliary data. However, the large discrepancies exist in different estimates. In this manuscript, the authors conducted a solid work by comparing different ET in European domain. This can help enhance understanding the spatial patterns of ET in Europe. I think the topic is quite nice, but I have some concern about the scientific presentations before it can be potentially published.
- I think the Fig. 1-5 can be re-organized as the authors already presented 5 figures in Materials, even some of them can be merged in one.
- Line 470-: I feel curious that there are large differences in two benchmark ET estimates. So how do you judge the RS-ET? In other words, why did you choose two benchmark estimates? Sometimes too many is too complicated.
- I feel confused in the whole results part, because I saw so many comparisons between different estimates, but the general core finding is not clear for me after reading. This suggests that the authors may consider clarifying the major information first before presenting.
- I think Fig.9 is repeated since it is just the normalized values, which make it different from the previous figures.
- Line 496-497: I think this is quite interesting finding. I would expect more discussions about this, but it was disappointing that I did not find...
- Line 651-654: I do not agree with the statement here.
The large differences in the ET estimates are caused by many factors. The forcing data, the formulations in estimating ET, as well as the spatial resolution.
Ensemble mean is not always a solution for the divergences in different estimates. The most important thing is to disentangle the factors (or at least, major factors) that cause the differences.
When I proceed my reading until here, I suddenly notice that I have not been told about the factors. So I recommend the authors to focus on the discussions about the factors causing the differences, instead of giving too many guesses. - Generally, I like the topic of this manuscript, but the presentation is not reader friendly because the authors wanted to cover everything but forgot to draw out the most interesting findings. I suggest the authors to refine or re-organize the manuscript with focus on the most important information. This would make this manuscript more interesting and benefit broader readers.
Author Response
Reviewer 2
ET is important for Earth's water and energy balance, despite the difficulties in accurate estimating ET. There are many new ET estimates with the emergence of long-term and high-resolution satellite data and auxiliary data. However, the large discrepancies exist in different estimates. In this manuscript, the authors conducted a solid work by comparing different ET in European domain. This can help enhance understanding the spatial patterns of ET in Europe. I think the topic is quite nice, but I have some concern about the scientific presentations before it can be potentially published.
Reply: We appreciate the generally positive review and very constructive comments.
- I think the Fig. 1-5 can be re-organized as the authors already presented 5 figures in Materials, even some of them can be merged in one.
Reply: The maps in figures 1, 2 and 4 could in principle be merged into one figure with five maps. However, these maps are introduced at different stages in the manuscript, so I don’t believe this would facilitate reading or the space they take up in the manuscript, it would just reduce the number of numbered figures. I don’t quite see how the Figures 3 and 5 could be merged with the other figures. I would really prefer to keep the figures in their current form, since I believe it facilitates reading.
- Line 470-: I feel curious that there are large differences in two benchmark ET estimates. So how do you judge the RS-ET? In other words, why did you choose two benchmark estimates? Sometimes too many is too complicated.
Reply: This is a valid point. Our hypothesis was that the benchmarks would be more similar. However, the analysis demonstrated the contrary. In our discussion we clearly question the WB-ET approach based on these gridded runoff data, and we do not trust this benchmark.Instead of taking out the WB-ET, we prefer to keep it and we will instead introduce the benchmarks with more description of the uncertainty. We deliberately chose the term benchmark instead of validation or “true” observations, to highlight that even the benchmarks are uncertain. By keeping both benchmarks, the study also sheads light on the challenged in establishing good benchmark or test datasets for evaluating large scale ET patterns. Corrections have been added to the introduction to the benchmarks (New lines 131-134). We feel that we are already expressing our distrust in the WB-ET in the current formulation e.g. “The WB-ET approach is obviously very sensitive to the discharge data used and the approach used here, based on gridded runoff produced by a machine learning approach seems to be inadequate”
- I feel confused in the whole results part, because I saw so many comparisons between different estimates, but the general core finding is not clear for me after reading. This suggests that the authors may consider clarifying the major information first before presenting.
Reply: Thanks for the comment. Part of the missing clarity in the result part can owe to the writing style of omitting most interpretation from the result section and leaving that for the discussion section, where we have a clearer description of the major information and findings. Also, In the conclusion we write:
“Four ET estimates (MODIS16, TSEB, PT-JPL and Budyko) display a high degree of agreement in spatial patterns across major river basins in continental Europe. Likewise, the within basin spatial patterns are analyzed across the moderate resolution RS-ET estimates revealing good agreement also at that scale especially between TSEB and PT-JPL and between TSEB and MODIS16, although with large variation between basins. These four estimates represent both purely energy constrained (MODIS16, TSEB and PT-JPL) as well as water and energy constrained (Budyko) methods. In addition, the RS-ET methods span algorithms driven mainly by vegetation (MODIS16) and driven by both vegetation and land surface temperature (TSEB and PT-JPL). This indicates that even with fundamental differences in methods similar spatial patterns in ET are obtained. A way forward could be to merge these four estimates, and other estimates that agree on the general patterns, in a robust ensemble.
The WB-ET estimate based on gridded runoff data based on a machine learning approach seems to be unsuited for at WB-ET approach, particularly high elevation regions. The resulting spatial pattern of the WB-ET method deviates significantly from the other estimates. The RS-ET estimate PML_V2 has a completely different ET pattern from all other estimates and does not follow the expected variability across Europe. Therefore PML_V2 seems unsuited as a reference for long-term spatial pattern of ET for Europe.”
This should be a quite clear summary of the important findings across all the analysis. All the analysis and results (visual, SPAEF, Budyko and Copulas) generally show this picture, that MODIS16, TSEB, PT-JPL and Budyko display a high degree of agreement in spatial patterns, and that WB-ET and PML_V2 are substantially different. This is also why we suggest and ensemble or merging approach based on the four estimates with high similarity. We have tried to highlight the major findings in the result section, (new lines 482-484 +541-545)..
- I think Fig.9 is repeated since it is just the normalized values, which make it different from the previous figures.
Reply: This is true, however with the emphasis on the spatial patterns, we really believe that the normalized maps facilitate interpretation. It could be argued that the non-normalized maps (Fig 7) are redundant, however since all statistics are spatial pattens statistics, Figure 7 is useful for the brief discussion of the discrepancies on actual ET levels between the estimates.
- Line 496-497: I think this is quite interesting finding. I would expect more discussions about this, but it was disappointing that I did not find...
Reply: Agreed, this discussion has been elaborated (new lines 503-511).
- Line 651-654: I do not agree with the statement here.
The large differences in the ET estimates are caused by many factors. The forcing data, the formulations in estimating ET, as well as the spatial resolution.
Ensemble mean is not always a solution for the divergences in different estimates. The most important thing is to disentangle the factors (or at least, major factors) that cause the differences.
When I proceed my reading until here, I suddenly notice that I have not been told about the factors. So I recommend the authors to focus on the discussions about the factors causing the differences, instead of giving too many guesses.
Reply: It is a valid point that ensembles are not always the best solution, in cases where model performance can be quantified systematically, and where the factors causing model differences can be identified and errors corrected. The argument for suggesting an ensemble approach in this case, is that at the continental scale we will not be able to determine which estimate is true and will have to accept this uncertainty, in such cases an ensemble approach can be useful (also highlighted by reviewer 3), and we do discard two of the estimates (WB-ET and PML_V2). However, the reviewer is right, that more understanding of what caused the differences would be advantageous. Our study does not analyze the detailed differences between the RS ET models, but approach the comparison from the resulting spatial patterns in ET. Therefore, discussion of the factors causing the dissimilarities will somewhat speculative. We believe, the major factors are 1) differences in the available energy (PET or Net radiation) and 2) the fundamental difference between temperature and vegetation-based estimates, 3) PML_V2 seems to be almost entirely driven by the North-South radiation gradient with little sensitivity to vegetation, elevation and other factors. 4) The pattern of the WB-ET is mainly controlled by the precipitation pattern. We have tried to expand the discussion of the factors controlling the discrepancies (new lines 618-620 + 643-646).
- Generally, I like the topic of this manuscript, but the presentation is not reader friendly because the authors wanted to cover everything but forgot to draw out the most interesting findings. I suggest the authors to refine or re-organize the manuscript with focus on the most important information. This would make this manuscript more interesting and benefit broader readers.
Reply: We have made adjustments that highlight the key findings and messages (new lines 482-484 +541-545). However, given the very positive reviews from the two other reviewers, we do not find it necessary to re-organize the manuscript.
Reviewer 3 Report
Overall a solid paper that is well focused and addresses important issues w/r to similarity of different RS and model based spatial patterns in actual evapotranspiration climatologies for Europe. The dis-similarities are at times disheartening, but the information is extremely important and the suggestion/conclusion that it should lead to ensembles is well founded. Some minor language edits are called for at places and improved resolution of figures 8 and 13 that appear as low quality scans and therefore somewhat blurry are needed.
Author Response
Reviewer 3
Overall a solid paper that is well focused and addresses important issues w/r to similarity of different RS and model based spatial patterns in actual evapotranspiration climatologies for Europe. The dis-similarities are at times disheartening, but the information is extremely important and the suggestion/conclusion that it should lead to ensembles is well founded. Some minor language edits are called for at places and improved resolution of figures 8 and 13 that appear as low quality scans and therefore somewhat blurry are needed.
Reply: Thanks for the positive review. We agree, that some of the dis-similarities are disheartening, but also find confidence in the general similarity between four of the ET-estimates. Figures 8 and 13 have been added in higher-resolution.